# Modern Hopfield Networks meet Encoded Neural Representations - Addressing Practical Considerations

**Satyananda Kashyap**
IBM Research - Almaden
satyananda.kashyap@ibm.com

**Niharika S. D'Souza**
IBM Research - Almaden

**Luyao Shi**
IBM Research - Almaden

**Ken C. L. Wong**
IBM Research - Almaden

**Hongzhi Wang**
IBM Research - Almaden

**Tanveer Syeda-Mahmood**
IBM Research - Almaden

## Abstract

Content-addressable memories such as Modern Hopfield Networks (MHN) have been studied as mathematical models of auto-association and storage/retrieval in the human declarative memory, yet their practical use for large-scale content storage faces challenges. Chief among them is the occurrence of meta-stable states, particularly when handling large amounts of high dimensional content. This paper introduces Hopfield Encoding Networks (HEN), a framework that integrates encoded neural representations into MHNs to improve pattern separability and reduce meta-stable states. We show that HEN can also be used for retrieval in the context of hetero association of images with natural language queries, thus removing the limitation of requiring access to partial content in the same domain. Experimental results demonstrate substantial reduction in meta-stable states and increased storage capacity while still enabling perfect recall of a significantly larger number of inputs advancing the practical utility of associative memory networks for real-world tasks.

## 1 Introduction

The hippocampal system in the brain plays a central role in long-term memory, responsible for storing and recalling facts and events. Its structure, particularly the auto-associative networks in the CA3 region, enables efficient memory retrieval based on partial input, a process that has inspired several mathematical models of memory networks (Almeida et al., 2007; Whittington et al., 2020; Gillett et al., 2020; Burns et al., 2022). Classical Hopfield network models (Hopfield, 1982; Amari, 1972) are a type of associative memory architecture that stores memories as fixed point attractor states in an energy landscape, using Hebbian learning to recall patterns form partial input cues using a recurrent network. The Modern Hopfield network (MHN) introduced a continuous relaxation of the original method and in theory enable exponential storage capacity growth with respect to the number of neurons (Krotov & Hopfield, 2016; Demircigil et al., 2017; Ramsauer et al., 2020). MHNs have been applied to tasks such as immune repertoire classification (Vaswani et al., 2017; Widrich et al., 2020) and graph anomaly detection (Hoover et al., 2023).

Extending these ideas, heteroassociative memories provide a framework for hippocampal memory storage and retrieval, enabling pattern recall from different input modalities. Hetero-association has also been interpreted in various ways, including for modeling sequence associations (Chaudhry et al., 2023; Karuvally et al., 2023; Gutfreund & Mezard, 1988), where the process begins with a given pattern in a sequence, adjusting the energy weights to transition from one pattern to the next. Similarly, (Tyulmankov et al., 2021) introduces a key-value memory model where sequential patterns are used to predict the next item in the sequence. Further, (Millidge et al., 2022) demonstrated

hetero-association by reconstructing missing portions of an image based on other parts of the same image. Most research on content addressable memories focus on dense associative memory theory, and use simplistic scenarios or simulations for validation (Kang & Toyoizumi, 2023; Iatropoulos et al., 2022). Translating to practical large-scale content storage systems has been difficult, as associative memories like MHNs tend to enter spurious meta-stable states when handling large volumes of content (Martins et al., 2023). The metastable states are closely related to the separability of the input patterns (Ramsauer et al., 2020). Additionally, MHNs rely on partial content for recall, which limits their utility in cases where such cues are unavailable.

In this paper we address the issue of meta-stable states in Modern Hopfield Networks (MHN) by improving the separability of input patterns through neural encoding representations. Our approach termed HEN (modern Hopfield networks with Encoded Neural representations or in short Hopfield Encoding Networks) encodes inputs into a latent representational space using a pre-trained neural encoder-decoder model before storage, and decodes them upon recall. While prior work has explored encoding content for associative memory (Kang & Toyoizumi, 2023), our method uniquely combines pre-encoding and post-decoding to specifically tackle metastable states in MHNs. Additionally, HEN supports hetero-association, allowing retrieval through free text queries, thus eliminating the need for users to provide partial content for recall. Comprehensive ablation experiments demonstrate that HEN significantly increases storage capacity, reduces metastable states across modalities, and enables perfect recall of a significantly larger number of stored elements using natural language input.

Our approach begins by surveying different representations of energy-based formulations, demonstrating how they form a unifying framework that connects support vector-based kernel memories (Iatropoulos et al., 2022), Modern Hopfield networks (Ramsauer et al., 2020; Krotov & Hopfield, 2020), and the transformer attention mechanism (Vaswani et al., 2017). By establishing these theoretical connections, we shed light on how similar representations emerge in distinct neural models.

Despite the theoretical promise of MHNs, we identify their practical limitations, notably the emergence of spurious states due to weak input pattern separability. **To address this issue, we propose HEN, which enhance pattern separation by encoding input representations into a latent space before storage.** This method delays the onset of metastability and significantly increases storage capacity, leading to improved recall stability compared to alternative representation learning strategies.

## 2 Modern Hopfield Networks: A Representational Perspective

In this section, we review the basic framework of Modern Hopfield Networks (MHNs), focusing on their representational properties, equivalence with other models, and inherent limitations. This discussion motivates our enhancements to MHNs using neural encoding strategies aimed at improving pattern separability and storage capacity.

The MHN framework provides a framework for dense associative memory using continuous dynamics. It can be described in terms of its energy function and the resulting attractor dynamics. Let $N$ be the number of memories and $K$ be the data dimensionality (number of neurons in the MHN). Defining a similarity metric between the memories $\{\xi_n \in \mathcal{R}^{K \times 1}\}_{n=1}^N$ or matrix $\Xi \in \mathcal{R}^{N \times K}$ and the state vector $\mathbf{s} \in \mathcal{R}^{K \times 1}$ the generalized objective function is expressed as the following energy minimization:

$$\mathbf{s}^* = \arg\min_{\mathbf{s}} E(\mathbf{s}, \Xi) = \arg\min_{\mathbf{s}} E_1(\mathbf{s}, \Xi; \beta) + E_2(\mathbf{s}) \tag{1}$$

$$E_1(\mathbf{s}, \Xi; \beta) = F_\beta(f_{\text{sim}}(\{\xi_n\}, \mathbf{s})) = F_\beta\Big(\{\sum_{k=1}^K \xi_n(k)\mathbf{s}(k)\}\Big) = -\frac{1}{\beta} \log \Big[ \sum_{n=1}^N \exp\Big(\beta \sum_{k=1}^K \xi(k)\mathbf{s}(k)\Big)\Big]$$

$$E_2(\mathbf{s}) = \frac{1}{2}\mathbf{s}^T\mathbf{s} + \text{constant}$$

The function $f_{\text{sim}}(\{\xi_n\}, \mathbf{s}; \beta)$ is a measure of similarity between the state vector and each memory in the bank. A common choice for this similarity metric is the dot product in $K$ dimensional vector space, which is both efficient and widely adapted in practical implementations of MHNs (Ramsauer et al., 2020; Krotov & Hopfield, 2020). The energy function $F_\beta(\cdot)$, defined as the log-sum-exponential (LSE), approximates the $\arg\max$ function to select the most relevant memory. The inverse temperature parameter $\beta$ controls the sharpness of this selection. The energy minimization

equation treats memories as attractors of a dynamical system (Krotov & Hopfield, 2016), recoverable from partial cues through an iterative optimization process. The state vector recurrence is given by:

$$\mathbf{s}^{(t+1)} = \mathbf{\Xi}\,\text{softmax}(\beta\mathbf{\Xi}^T\mathbf{s}^{(t)}) = \frac{\sum_n \xi_n \exp(\beta\xi_n^T\mathbf{s}^{(t)})}{\sum_n \exp(\beta\xi_n^T\mathbf{s}^{(t)})} \tag{2}$$

These updates progressively reduce the energy of the system monotonically (Millidge et al., 2022) and are guaranteed to converge under specific conditions (Ramsauer et al., 2020). Starting with an initial state $\mathbf{s}^{(0)}$, which could be a partial or noisy memory cue, the iterations $\{\mathbf{s}^{(t)}\}$ aims to reconstruct a full pattern $\mathbf{s}^{(T_f)}$ that corresponds to one of the stored memories $\{\xi_n\}$. However, in practice, the process may lead to local minima or saddle points, resulting in meta-stable configurations.

## 2.1 Equivalence with Kernel Memory Networks (KMN)

The formulation of the MHNs can be viewed as a special case of Kernel Memory Networks (KMNs) (Iatropoulos et al., 2022), where each neuron performs kernel-based classification or regression. The MHN update rule is analogous to that in KMNs and can be computed in closed form as a recurrence. Let $K(\mathbf{\Xi}, \mathbf{s}^{(t)})$ denote the (symmetric positive definite) kernel function that defines pairwise similarities, and $\mathbf{K}^\dagger$ be the Moore-Penrose pseudoinverse for $\mathbf{K}(\mathbf{\Xi}, \mathbf{\Xi})$. For continuous valued memories, the radial translation-invariant exponential kernel (infinite dimensional basis) with a fixed spatial scale $r$ and temperature $\alpha$ is proposed. This parameterization allows for an analysis of memory capacity and storage limits by interpreting the MHN optimization as a feature transformation operating in a Reproducing Kernel Hilbert Space (RKHS). The kernel and state update are as follows:

$$\text{Kernel: } K_{(\alpha,r)}(\mathbf{x},\mathbf{y}) = \exp\left[-\left(\frac{1}{r}||\mathbf{x}-\mathbf{y}||_2\right)^\alpha\right] \quad \text{State Update: } \mathbf{s}^{(t+1)} = \mathbf{\Xi}\mathbf{K}^\dagger K(\mathbf{\Xi}, \mathbf{s}^{(t)}) \tag{3}$$

While KMNs provide strong theoretical storage guarantees (Iatropoulos et al., 2022), their real-world performance is known to be sensitive to data distributions and parameter choices Wu et al. (2024).

## 2.2 Equivalence with Transformers

Alternatively, the update rule in Eq. (2) has been shown (Ramsauer et al., 2020) to be equivalent to the key-query *self*-attentional framework used in transformer models (Vaswani et al., 2017).

$$\mathbf{Z} = \text{softmax}\left(\frac{1}{\sqrt{d_k}}\mathbf{Q}\mathbf{K}^T\right)\mathbf{V} \quad \text{or} \quad \mathbf{Z}^T = \mathbf{V}^T\text{softmax}\left(\frac{1}{\sqrt{d_k}}\mathbf{K}\mathbf{Q}^T\right) \tag{4}$$

with the keys being related to the memories as $\mathbf{W}_K\mathbf{\Xi} = \mathbf{K}$, and queries/values being related to the intermediate state vectors $\mathbf{s}^{t+1} = \mathbf{Z}^T$, $\mathbf{W}_Q\mathbf{s}^t = \mathbf{Q}^T$, $\mathbf{W}_V\mathbf{s}^t = \mathbf{V}^T$ and the dispersion parameter relating to the temperature $\frac{1}{\sqrt{d_k}} = \beta$. The matrices $\mathbf{W}_Q, \mathbf{W}_K, \mathbf{W}_V : \mathcal{R}^K \to \mathcal{R}^D$ are linear transformations associated with the query-key-value triplet, which when substituted with the identity matrix $\mathcal{I}_K$ gives us the form in Eq. (2)

Under the kernel memory networks framework, the MHN equations (1-2,4) do not involve a symmetric positive definite kernel, as the energy objective is inherently non-convex (Iatropoulos et al., 2022; Wright & Gonzalez, 2021). However, the system still permits a bilinear reproducing form for the kernel $K(\cdot, \cdot)$, where input patterns are mapped into higher-dimensional feature spaces. This bilinear kernel is equivalent to the transformer's key-query attention mechanism, where inputs are projected into higher dimensional feature spaces. The transformer kernel can be written as: $K(\mathbf{x},\mathbf{y}) = \exp\left[\frac{1}{\sqrt{d_k}}(\mathbf{W}_Q\mathbf{x})^T(\mathbf{W}_K\mathbf{y})\right]$. Overall, this highlights the connection between the MHNs and transformer attention mechanisms, showing that both rely on projecting input representations into a shared space for similarity-based comparison. Further details on the KMN and transformer equivalence are provided in Appendix Section 1.

## 3 HEN: Modern Hopfield Networks with Encoded Neural Representations

Although the results of modern Hopfield networks and its various equivalent forms imply that the formulation has theoretically exponential capacity to store memories, *our experimental results*

*demonstrate that the system of updates can be brittle in practice and highly sensitive to real-world data distributions across each of these data representations.*

Specifically, MHN often struggle with spurious attractor basins (Bruck & Roychowdhury, 1990; Ramsauer et al., 2020; Barra et al., 2018), which manifest as erroneous memory patterns due to overlapping or similar inputs. This issue is particularly evident in large datasets, where poor pattern separability results in meta-stable states, limiting retrieval accuracy and scalability. The key insight from the unified representations discussed in Sections 2.1 and 2.2 is that improving the separability of input memories significantly enhances retrieval accuracy. This can be achieved by mapping input memories $\mathbf{\Xi}$ and partial queries $\mathbf{s}^{(0)}$ from their original $K$-dimensional space (where memories may overlap or be less distinct) into a higher dimensional embedding space, the stored patterns become better separable. The increased separability directly reduces the occurrence of spurious attractors and leads to more reliable memory retrieval.

Going one step further, a key idea we put forward here is to see if we can bolster the separability of the input patterns before they enter the Modern Hopfield network in order to reduce the spurious attractor states problem via large pre-trained encoded-decoder models and their latent space representations (i.e. generalizing the linear transforms in Eq. 4). Following observations of the phenomenon of input encoding in the dentate gyrus (DG) region of the trisynaptic circuit prior to memorization (Bernier et al., 2017), we propose to store these latent-space neural encodings, i.e. transformation computed on the memories $\hat{\mathbf{\Xi}}$ and partial query $\hat{\mathbf{s}}^{(0)}$, in the MHN memory bank using the encoder transformation $\mathbf{\Phi}_{\text{enc}}(\cdot)$. Recovery of such patterns can be performed by unrolling the recurrence relation in the latent-space (i.e. Eq.(2)) followed by applying the associated decoder transformation $\mathbf{\Phi}_{\text{dec}}(\cdot)$ to the latent space representation $\hat{\mathbf{s}}^{(T_f)}$. Mathematically, this procedure can be expressed as follows:

$$\hat{\mathbf{\Xi}} = \mathbf{\Phi}_{\text{enc}}(\mathbf{\Xi}) \quad \hat{\mathbf{s}}^{(0)} = \mathbf{\Phi}_{\text{enc}}(\mathbf{s}^{(0)}) \tag{5}$$

$$\hat{\mathbf{s}}^{(t+1)} = \hat{\mathbf{\Xi}}\text{softmax}(\beta\hat{\mathbf{\Xi}}^T\hat{\mathbf{s}}^{(t)}) = \frac{\sum_n \hat{\xi}_n \exp(\beta\hat{\xi}_n^T\hat{\mathbf{s}}^{(t)})}{\sum_n \exp(\beta\hat{\xi}_n^T\hat{\mathbf{s}}^{(t)})} \tag{6}$$

$$\mathbf{s}^{(T_f)} = \mathbf{\Phi}_{\text{dec}}(\hat{\mathbf{s}}^{(T_f)}) \tag{7}$$

*Hypothesis 1: The spurious attractors can be reduced by encoding inputs prior to storing them in the Modern Hopfield network and decoding them after recall due to increased separability in latent space*

Our proposed HEN combines an auto-encoder with the Modern Hopfield network (MHN). Specifically, the encodings are generated by a pre-trained auto-encoder. The raw content is then recovered through chaining MHN with the decoder portion of the auto-encoder. We hypothesize that the encodings produced by an auto-encoder contain discriminative information that is not only compact but can improve the separability in the energy landscape to significantly delay the meta-stable states even with increased content. That is, by leveraging a well-trained auto-encoder for feature extraction, we posit that the most significant and discernible features between images can be easily identified, leading to less spurious patterns emerging during recall and allowing more content to be stored, thereby increasing storage capacity.

### 3.1 Experimental Evaluation and Results:

We provide practical insights drawn from experiments on the MS-COCO dataset, which contains 110,000 images (Lin et al., 2015). This dataset offers a more realistic distribution of high-dimensional, real-world data compared to smaller, curated datasets like MNIST. Its unique associative captions also make it ideal for illustrating hetero-associations within our proposed framework.

To evaluate this hypothesis, we conducted studies that examined the effectiveness of various pre-trained encoder-decoder architectures to produce encoded representations that can lead to successful recall of dense associative memories by comparing them against the native data representations and KMNs. We also analyzed the parameter choices for the energy formulation of MHNs in affecting the identity of the recalled memory items when using their encoded representations.

Specifically, we evaluated various pre-trained encoder-decoder architectures known for their state-of-the-art performance in deep learning-based image encoding and decoding ranging from vanilla-transformer based to variational auto-encoder based models. In particular, we utilized the Discrete Variational Autoencoder (D-VAE) from (Ramesh et al., 2021) and other architectures from (Rombach et al., 2021) and explored two transformer variants from the diffusion library: one trained

with codebook-based (Vector Quantized - VQ) criteria and the other using Kullback-Leibler (KL) divergence-based criteria. Our empirical analysis revealed that Vector Quantized VAE (VQ-VAE) methods outperformed others in our setup. Consequently, we selected D-VAE and variants of VQ from (Rombach et al., 2021) for further analysis. To maintain consistency in representation, we downsampled all the images to a resolution of $28 \times 28 \times 3$ to match the number of features that the encoded representations produced.

**Evaluation Metrics**: This study tested the image-based dense MHNs (Eq. 1) and KMNs with the exponential kernel against pre-trained Discrete VAE (Ramesh et al., 2021) and VQ-VAEs encoding equipped Hopfield encoding network (Rombach et al., 2021) (Eq. 7). The test was conducted on a memory bank ($\{\xi_n \in \mathcal{R}^{1 \times K}\}$) and query size ($N$) of 6000 images from the MS-COCO dataset. To examine the effect of different choices of $f_{\text{sim}}(\cdot, \cdot)$ in Eq. (1), both dot product and $\ell_2$ based similarity measures were utilized. The performance of different encoder-decoder architectures was evaluated (see Fig. 2) by varying the dimensionality $K$. The Mean Squared Error ($MSE$) and Structural Similarity Index ($1 - SSIM$) metrics were used to compute the similarities between the encoder reconstructions stored in the $\{\xi_n\}$ memory bank and the reconstructed ones.

Note that as this evaluation was conducted to assess the performance on metastable states, *the focus was on recovering the correct identity rather than the quality of reconstruction*. Hence, a $MSE = 1 - SSIM = 0$ indicated that the dense associative memory could retrieve the full encoded representation of the image from which the pre-trained decoder could reconstruct the image. [1]

**Results:** Figure 2 shows the result of our analysis using six different encoding methods including raw image store and two similarity types (dot product and $\ell_2$). The encoded representations uniformly perform well above a certain $\beta$ value. In comparison, we note that the baseline image-based MHN (Dot-image & L2-image) persists in meta-stable states irrespective of the $\beta$ value or the similarity metric. Additionally, we also assessed metastable states by recording the relative reduction in the rank of the update matrix with the collapse of the pattern recovery process. The result is available in the Appendix, Section 2 and shows that for a judiciously chosen value of $\beta$, the iterates of HEN stabilize to provide near perfect retrieval, consistent with the trends observed in Figure 2.

**KMNs Sensitivity to Real-World Conditions**: KMNs have strong theoretical guarantees, but in our experiments, they faced significant challenges with real-world, large-scale image retrieval tasks. Despite an extensive parameter sweep over hyperparameters like $r$ (spatial scale) and $\alpha$ (inverse temperature), KMNs were highly sensitive to these settings and consistently underperformed. For

---

[1] We run a sanity check based on the column rank of the recovered pattern matrix $\hat{\mathbf{S}}^{(T_f)}$ to illustrate that these metrics correlate well with degeneracy in the recovered patterns. See Section 3 in the Appendix.

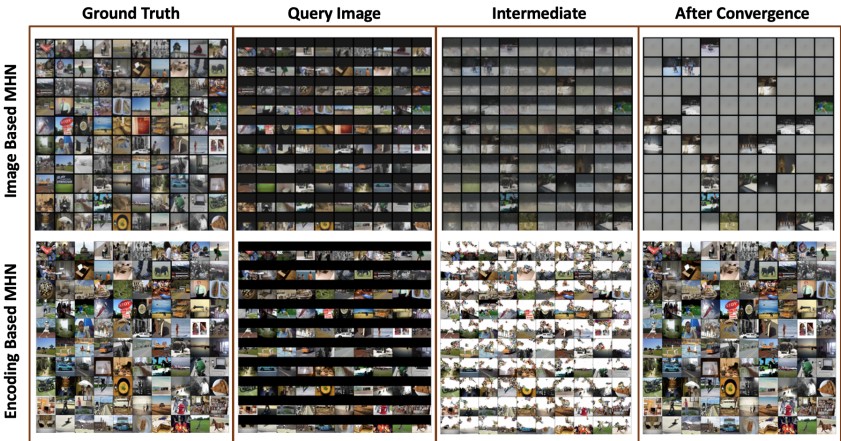

Figure 1: Progression of the **(Top Row)** Modern Hopfield Network (MHN) run on image inputs and the **(Bottom Row)** HEN at different intermediate steps. The sequence from left to right is as follows: the original image, the query image with half of it occluded, an intermediate update at iteration 11, and the final reconstruction at iteration 150. This uses an $\ell_2$ similarity and discrete Variational Autoencoder (D-VAE) encoder for the HEN. We set $\beta = 150$ in both experiments

Table 1: All neural encoders can recover images without quality loss as more data is stored in $\{\xi_n\}$.

| NUM IMAGES $1 - SSIM, MSE$ | vqf8 | vqf16 | Image | D-VAE |
|---|---|---|---|---|
| 6000 | 0.021, 0.000 | 0.019, 0.004 | 0.836, 0.064 | **0.000, 0.000** |
| 8000 | 0.019, 0.000 | 0.046, 0.004 | 0.835, 0.067 | **0.000, 0.000** |
| 10000 | 0.019, 0.000 | 0.047, 0.004 | 0.835, 0.064 | **0.000, 0.000** |
| 15000 | 0.019, 0.000 | 0.048, 0.004 | 0.836, 0.066 | **0.000, 0.000** |

example, with a memory bank of 6000 images, KMNs yielded an MSE of 0.2427 and 1-SSIM of 0.972, indicating poor retrieval accuracy. While KMNs excel in controlled environments, their performance degrades in high-dimensional, non-linearly separable data, limiting their practical use. This is suspected to be a direct consequence of the restrictive assumptions made on the underlying data-distributions in KMNs (Wu et al., 2024). In contrast, our HEN shows more robust performance, mitigating meta-stable states and improving retrieval accuracy.

**Quality of recovery**: Fig. 1 shows the result of using D-VAE encoding for perfect memory recall for the same set of images for which raw image storage in the Hopfield network failed. While the reconstruction quality is not as clear as the original, the identity of the recovered images is preserved one-to-one. In comparison, the recall using the raw images for the same dataset using the Modern Hopfield network shows the metastable states.

**Scale-out Performance and Ablation Study on Encoder-Decoder Pairs**: We evaluated HEN's scalability by testing it's retrieval performance as the number of stored images increased in the memory bank. Table 1 presents the performance metrics across different encoder-decoder approaches as the memory bank scaled from 6,000 to 15,000 images. Our results show that all the encoder-based approaches robustly recovered the image representations without a noticeable drop in the reconstruction quality, supporting HEN's stability and scalability in retrieval performance at this scale. Additionally, our ablation studies included five pre-trained encoder-decoder architectures - dVAE (Ramesh et al., 2021) and VQ-F8, VQ-F16, KL-F8, KL-F16 from (Rombach et al., 2021) to assess whether HEN's retrieval stability holds across varied configurations. Section 2 of the Appendix contain a more detailed breakdown of the recovery performance as a function of encoder-decoder methods, and the similarity metric for different values of $\beta$. *By contrast, KMNs consistently failed to scale with the increase in images, showing degraded performance as the number of images increased.*

**Probing the separability of HEN encodings:** To study separability of various encoding strategies, we examine the strength of association patterns in the HEN memory bank, i.e. the latent-space vectors. Extending the notation in Section 3, let $\hat{\mathbf{S}}_i^{(0)} \in \mathcal{R}^{K \times 1} = \mathbf{\Phi}_{\text{enc}}(\mathbf{S}_i^{(0)})$ denote the encoded query for example $i$ in the dataset. This encoding is generated by occluding a portion of the image fed

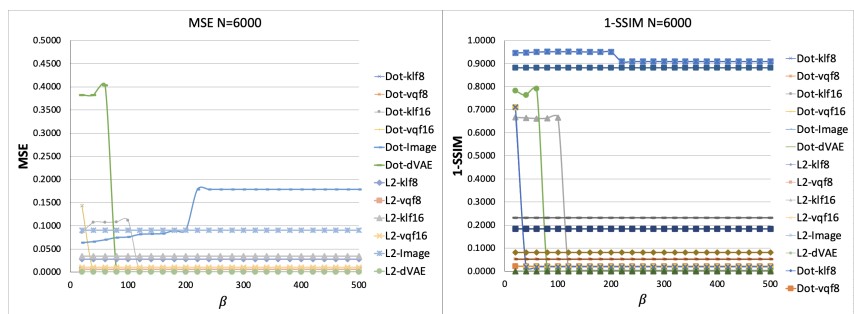

Figure 2: Memory recall performance of various encoder methods and the image-based Modern Hopfield network, each color-coded differently. The **(Left)** figure plots the MSE while the **(Right)** depicts the 1-SSIM as a function of $\beta$. The encoder-based HEN methods outperform the raw image-based method over a very large range of choices of hyperparameters. Dot or L2 in the legend denote the dot product or $\ell_2$, followed by the type of representation used including Original input-Image, dVAE-Discrete Variational Auto Encoder, Kullback-Leibler (KL)-based variants- KLf8, KLf16, and Vector quantized VAE methods-VQf8, VQf16 per the convention in (Rombach et al., 2021)

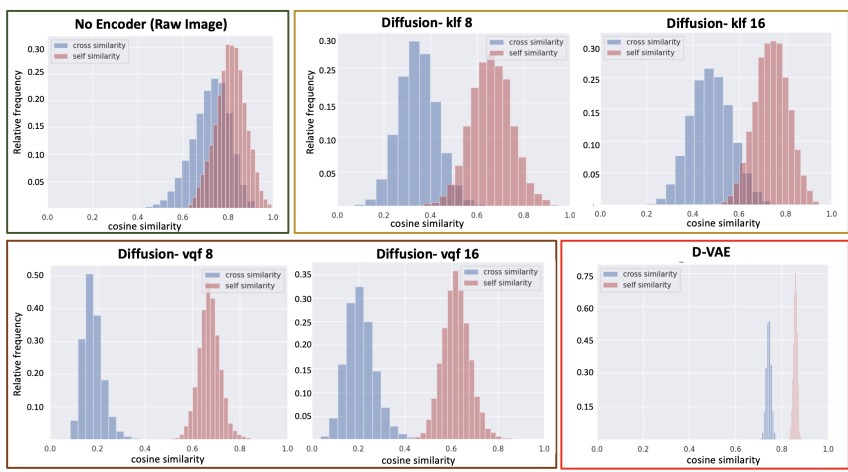

Figure 3: Illustrating separability in various embeddings used in Subsection 3. We plot the histogram of the distribution of cosine similarity values between the queries and memories, i.e. $\cos(\hat{\mathbf{S}}_i^{(0)}, \hat{\xi}_j)$. Distributions colored indicate self-similarity (i.e. $i = j$) across paired examples, while distributions colored in blue indicate cross similarities (i.e. $i \neq j$). We generate these distributions for (a) Raw Images in the Black Box, Diffusion models trained (b) on KL Divergence in the Orange Box, (c) trained using Vector Quantization in the Brown Box, and (d) Discrete-VAE (D-VAE) in the Red Box.

through the encoder (or just the occluded image for the raw image MHN). We expect that the major contributor to poor recovery performance is the lack of separation between the attractor basins in in Eq. (1), due to which the dynamics of state evolution $\hat{\mathbf{S}}_i^{(t)}$ in Eq. (2) are meta-stable configurations. To quantify this separation, we compute the cosine similarity between pairs of query and memory vectors, i.e. $c_{ij} = \cos(\hat{\mathbf{S}}_i^{(0)}, \hat{\xi}_j) = \hat{\xi}_j^T \hat{\mathbf{S}}_i^{(0)}/||\hat{\mathbf{S}}_i^{(0)}||_2||\hat{\xi}_j||_2$. If the patterns are well separated, each query $\hat{\mathbf{S}}_i$ in the encoding space (or in the native space for raw images) is close to its own memory $\hat{\xi}_i$ but far apart from others $\hat{\xi}_j, \forall j \neq i$. We test this in Fig. 3 by plotting the distribution of values as histograms for $c_{ij} \forall j \neq i$ colored in blue and for $c_{ii}$ colored in red, for different $\mathbf{\Phi}_{\text{enc}}(\cdot)$s in HEN.

This separation is poor for the raw image case, with the two histograms having a high overlap in values. This overlap substantially reduces across the neural encoder-based models, with the Vector Quantized variants providing improved separability and a relatively higher magnitude of self-similarity values compared to their KL counterparts. Finally, we notice that the D-VAE encoder, besides providing separable encodings, also results in the tightest fit around the mean for the self and cross-similarity value distributions. This is likely why the D-VAE provided the best performance (Fig. 2 and Table 1).

## 3.2 Natural language-based Hetero-associations

We now extend the HEN framework in a hetero-associative setting as a practical application. Specifically, we explore the use of cross-stimuli coming from language and vision, as language-based queries are often used as cues for recall, for example, in practical storage/retrieval contexts. Cross-associative features have been previously demonstrated for the classical Hopfield networks model, albeit under the limited setting of carefully curated binary patterns (Shriwas et al., 2019).

*Hypothesis 2: Hopfield encoding networks serve as content-addressable memories even with cross-stimuli associations as long as they are unique associations.*

We conducted three separate experiments. First, we use a native textual embedding for the language cue and associate it with the content to be stored as a practical way of enabling recall. Next, we explore the paradigm of stimuli type-conversion to render the language cue into a convenient image form to allow for content-based access. Finally, we show that if the uniqueness of association is lost, spurious memory states could again emerge even if the inputs are encoded.

Fig. 4 illustrates the overall methodology for the Hopfield Encoding Network (HEN) under hetero-associations. Here, the memory bank vectors are formed by concatenating image and text embeddings

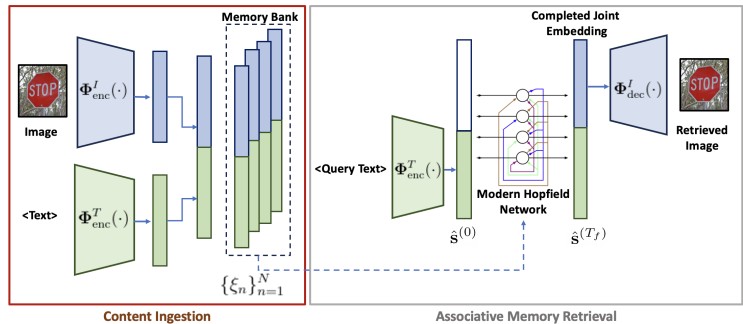

Figure 4: HEN architecture for Natural language based hetero-associations. **Orange Box:** Paired text and image inputs are fed to text ($\mathbf{\Phi}^{\mathbf{T}}_{\text{enc}}(\cdot)$) and image ($\mathbf{\Phi}^{\mathbf{I}}_{\text{enc}}(\cdot)$) encoders respectively to generate the memories of the HEN memory bank. **Grey Box:** At query time, a partial query $\hat{s}^{(0)}$ is generated by feeding the query text into $\mathbf{\Phi}^{\mathbf{T}}_{\text{enc}}(\cdot)$. After convergence, the image encoding is extracted from $\hat{s}^{(T_f)}$, and decoded through the Image decoder ($\mathbf{\Phi}^{\mathbf{I}}_{\text{dec}}(\cdot)$) to retrieve the corresponding image. Using full image representations instead of image encodings implies $\mathbf{\Phi}^{\mathbf{I}}_{\text{enc}}(\cdot) = \mathbf{\Phi}^{\mathbf{I}}_{\text{dec}}(\cdot) = \mathcal{I}_K$, the identity transformation. In the experiment where the text captions are pixelized as input, $\mathbf{\Phi}^{\mathbf{T}}_{\text{enc}}(\cdot) = \mathbf{\Phi}^{\mathbf{I}}_{\text{enc}}(\cdot)$.

Table 2: Performance of encoded cross-modal HEN compared to image-based MHNs as the memory bank $\{\hat{\xi}_n\}$ increases. The first row shows the CLIP-encoded cross-modal representations, while the rest present pixelized text-encoded representations for increasing dataset sizes.

| NUM IMAGES 1-SSIM, MSE | vqf8 | vqf16 | Image | D-VAE |
|---|---|---|---|---|
| 6000-CLIP | 0.016, 0.000 | 0.024, 0.000 | 0.681, 0.118 | **0.000, 0.000** |
| 6000 | 0.016, 0.000 | 0.024, 0.000 | 0.952, 0.214 | **0.000, 0.000** |
| 8000 | 0.016, 0.000 | 0.023, 0.000 | 0.952, 0.215 | **0.000, 0.000** |
| 10000 | 0.015, 0.000 | 0.024, 0.000 | 0.952, 0.215 | **0.000, 0.000** |
| 15000 | 0.015, 0.000 | 0.024, 0.000 | 0.952, 0.215 | **0.000, 0.000** |

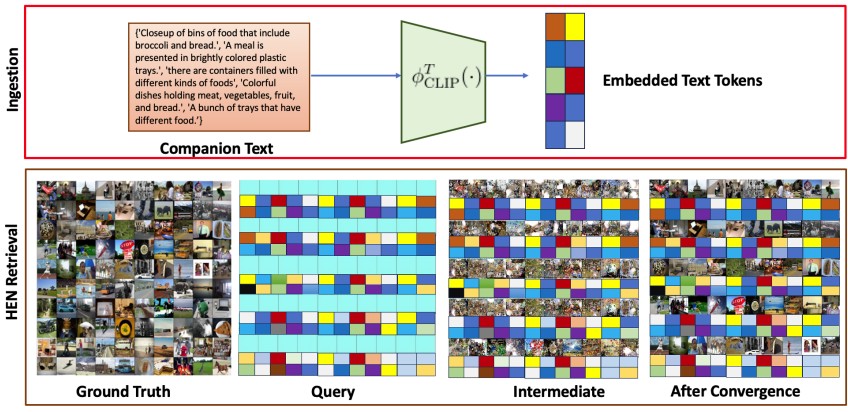

Figure 5: Step-by-step progression of a cross-modal query using two different encodings for associated visual and language cues. **Top Row:** The text stimulus is encoded via the CLIP (Radford et al., 2021) text encoder and associated with the image represented by a D-VAE encoded vector. **Bottom Row:** The reconstruction process for heteroassociation. **(L-R)** Ground Truth, Iteration $t = 0$ starting with a blank canvas with the provided CLIP Encoded text inputs as query prompts, an intermediate update, and the full reconstruction. This demonstrates the network's ability to accurately reconstruct the image from a text-only input from a completely different stimulus space as the image content.

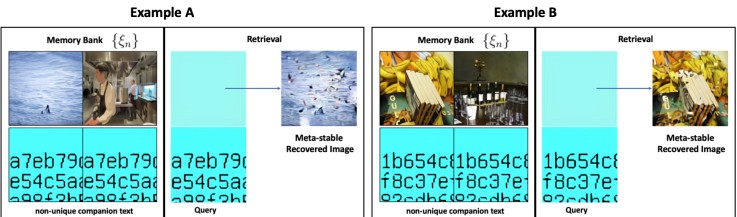

Figure 6: In each example, **(Left)** the upper and lower image cases depict the disruption of unique associations in the $\{\hat{\xi}_n\}$ memory bank. **(Right)** A single text input corresponding to these disrupted associations is used during the query phase. The top blue image represents an empty image as a zero-encoded vector at initialization. The reconstruction appears to be a meta-stable state. Neither of the original images is accurately recovered, supporting the hypothesis of unique text-image associations.

$\hat{\xi}_n = [\boldsymbol{\Phi}^{\mathbf{I}}_{\text{enc}}(\mathbf{I}_n); \boldsymbol{\Phi}^{\mathbf{T}}_{\text{enc}}(\mathbf{T}_n)]$. During retrieval, we construct a query vector $\hat{\mathbf{s}}^{(0)} = [\mathbf{0}; \hat{\mathbf{s}}_{\mathbf{T}}]$ constructed using the encoded text vector $\hat{\mathbf{s}}_T = \boldsymbol{\Phi}^{\mathbf{T}}_{\text{enc}}(\mathbf{s}^{(0)}_{\mathbf{T}})$ and zeros in the location of the image encodings. Finally, after convergence $\hat{\mathbf{s}}^{(T_f)} = [\hat{\mathbf{s}}^{(T_f)}_{\mathbf{I}}; \hat{\mathbf{s}}^{(T_f)}_{\mathbf{T}}]$, we decode the image embedding $\mathbf{s}^{(T_f)}_{\mathbf{T}} = \boldsymbol{\Phi}^{\mathbf{I}}_{\text{dec}}(\hat{\mathbf{s}}^{(T_f)}_{\mathbf{I}})$ to retrieve the image content.

To test language-image associations, we utilized the unique set of captions associated with each image in the COCO dataset. In our experiment, we allowed image and text to be encoded with different encoder decoder architectures. Specifically, we retained the best performing encoder (D-VAE) for image encoding (See Fig. 5) but the textual associative stimulus was encoded using the CLIP foundational model (Radford et al., 2021).

To create a more meaningful embedding, we concatenated the set of caption sentences per image into a single long sentence. This sentence was then encoded using the pre-trained CLIP model. The resulting text and image encodings were then ingested into HEN, as illustrated in Fig. 5 (Top). The experiment yielded promising results. The performance of the 6,000 images tested was on par with that of the discrete VAE in the same embedding space. *We found this to be significant, as it suggests that text and image encoders can operate in disjoint spaces while still achieving accurate reconstructions*, provided the Hopfield energy landscapes are appropriately normalized. Further, the top row of Table. 2 shows robust performance across all CLIP combinations.

We also explored an alternate strategy for representing cross-stimulus cues in a single input space by pixelizing the text representations into a unique 'image' representation (See example in Fig. 6) instead of a text-encoder. Converting this text into an image allows us to re-use the image encoder-decoder for both image and text portions of the queries and memory bank latent-space transformations. To our surprise, this schema provided comparable recovery performance to using dedicated image and text encoding strategies in Table. 2. See Section 4 in the Appendix for details on the pixelization.

Finally, to examine the uniqueness of association hypothesis, we designed an experiment in which two different images to be stored in HEN were selected at random and associated with the same textual pattern. We rendered the text in a pixelized form and used the same encoding as for image to remove the effect of separate encodings for image and text in testing the uniqueness aspect. We queried the system using the pixelized text to observe the type of images that would be reconstructed. Fig. 6 indicates that violating the uniqueness constraint led to spurious recall where the reconstructed image appeared to be a mixture of two different images. Thus HEN can support cross-stimuli associations and the recall is accurate if the associative text pattern is distinct per image.

## 4   Conclusions

In this paper, we unified and explored the diverse energy formulations used across associative memory models, highlighting their theoretical interconnections and the shared focus on pattern separability. This foundational analysis informed our development of key enhancements for Modern Hopfield Networks (MHNs), specifically by integrating pattern encoders and decoders to enhance separability and reduce metastable states. Additionally, we demonstrated how this approach supports cross-stimuli associations using different encodings, as long as the uniqueness of association is maintained

suggesting promising adaptability for applications that require cross-modal recall, like multimedia search or multi-sensor data fusion. These advancements mark a step toward improving the practicality and performance of Modern Hopfield Networks for real-world retrieval and storage tasks.

Additionally, HEN's architecture shows potential for other associative tasks, such as temporal sequences or visual variations (e.g., images of the same entity from different angles). Prior studies on associative memory Gutfreund & Mezard (1988); Chaudhry et al. (2023); Shriwas et al. (2019); Millidge et al. (2022) suggest that with minor modifications, HEN could be extended to sequential or contextual retrieval, supporting broader applications in dynamic and context-rich environments.

While HEN has shown consistent performance, we recognize that its convergence and stability depend on factors such as the update rule, encoder-decoder selection, and time step $T_f$. Our results indicate that after a set number of iterations ($T_f = 100$ in our case), retrieval dynamics reach a stable plateau; however, the optimal $T_f$ may vary by dataset. Thus, while our approach delays metastable states, further tuning may be necessary in different contexts.

Looking forward, future work could investigate HEN's scalability with even larger datasets by implementing fine-tuning steps to optimize VAE parameters, potentially further reducing metastable behavior. This approach could enhance HEN's memory capacity, enabling it to support practical storage and retrieval use cases at scale. These extensions position HEN as a versatile framework, adaptable to diverse associative memory tasks and scalable to meet the demands of large-scale, heterogeneous data environments.

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
