# [Appendix] Modern Hopfield Networks meet Encoded Neural Representations - Addressing Practical Considerations

**Satyananda Kashyap**
IBM Research - Almaden
satyananda.kashyap@ibm.com

**Niharika S. D'Souza**
IBM Research - Almaden

**Luyao Shi**
IBM Research - Almaden

**Ken C. L. Wong**
IBM Research - Almaden

**Hongzhi Wang**
IBM Research - Almaden

**Tanveer Syeda-Mahmood**
IBM Research - Almaden

## 1 Transformers and Kernel Memory Networks Representations

From a representational perspective, (Iatropoulos et al., 2022) demonstrates that the formulations in Eqs.(1-2) in the main manuscript are special instances of Kernel Memory Networks (KMN), i.e. kernel machines from statistical learning, that train each individual neuron in the MHN to perform either kernel classification or interpolation (regression) with a minimum weight norm. Under this specialization, the update rule for an auto-associative memory framework (such as the classical Hopfield or Modern Hopfield Networks) can be expressed in a form analogous to Eq. (2) in the main manuscript:

$$\mathbf{s}^{(t+1)} = \mathbf{\Xi} \mathbf{K}^{\dagger} K(\mathbf{\Xi}, \mathbf{s}^{(t)}), \tag{1}$$

where the kernel function $K(\cdot, \cdot) : \mathcal{R}^K \times \mathcal{R}^K \to \mathcal{R}$ is defined pairwise between two input vectors in K-dimensional space, and $\mathbf{K} = K(\mathbf{\Xi}, \mathbf{\Xi}) = \phi(\mathbf{\Xi})^T \phi(\mathbf{\Xi})$ is the kernel matrix associated with the feature transformation $\phi(\cdot) : \mathcal{R}^K \to \mathcal{R}^D$ to a D dimensional vector space. $\mathbf{K}^{\dagger}$ is its Moore-Penrose pseudoinverse, where $\mathbf{K}^{\dagger} = \mathbf{K}^{-1}$ if $\phi(\mathbf{\Xi})$ is full column rank.

This framework is applicable under the restriction that $K(\cdot, \cdot)$ is a Mercer kernel (symmetric-positive definite), admits a reproducing bilinear form where the transformation $\mathbf{\Phi}(\cdot)$ is a reproducing kernel hilbert space (RKHS). Under these restrictions, (Iatropoulos et al., 2022) lays the groundwork to analyze the storage capacity and corresponding recovery guarantees for query patterns (i.e. proximity to attractor basins) and their relation to the properties of the kernel function and the optimization procedure. In the case of continuous valued memories (non-binary case), a standard choice of a radial translation-invariant exponential kernel (with infinite dimensional basis) with a fixed spatial scale $r$ and temperature parameter $\alpha$.

$$K_{(\alpha, r)}(\mathbf{x}, \mathbf{y}) = \exp\left[ -\left(\frac{1}{r}||\mathbf{x} - \mathbf{y}||_2\right)^{\alpha}\right] \tag{2}$$

The KMN optimization is expected to be well-posed, convex, and is expected to provide global convergence to one of the stored memories (i.e. no meta-stable outputs) in theory.

Preprint.

## 1.1 Relating Transformer and HEN Updates to the KMN formulation:

Under the KMN formulation, the MHN form involves a kernel that is not symmetric positive definite, as the energy objective is not a convex optimization problem (Iatropoulos et al., 2022; Wright & Gonzalez, 2021). Nevertheless, it permits a bilinear reproducing form for the kernel $K(\cdot, \cdot)$, where the feature transformation maps $\{\boldsymbol{\Phi}_{\mathcal{X}}(\cdot) : \mathcal{X} \to \mathcal{F}_{\mathcal{X}}(\cdot), \boldsymbol{\Phi}_{\mathcal{Y}}(\cdot) : \mathcal{Y} \to \mathcal{F}_{\mathcal{Y}}\}$ map input elements from vector spaces $\{\mathbf{x} \in \mathcal{X}, \mathbf{y} \in \mathcal{Y}\}$ to the D-dimensional Banach spaces $\{\boldsymbol{\Phi}_{\mathcal{X}}(\cdot) \in \mathcal{F}_{\mathcal{X}}, \boldsymbol{\Phi}_{\mathcal{Y}}(\cdot) \in \mathcal{F}_{\mathcal{Y}}\}$, characterized by linear manifold functional forms $\{\mathbf{f}_{\boldsymbol{\Phi}_{\mathcal{X}}}(\mathbf{x}; \mathbf{W}_Q) \in \mathcal{B}_{\mathcal{X}}, \mathbf{g}_{\boldsymbol{\Phi}_{\mathcal{Y}}}(\mathbf{y}; \mathbf{W}_K) \in \mathcal{B}_{\mathcal{Y}}\}$. The mathematical expression for the transformer kernel is:

$$K(\mathbf{x}, \mathbf{y}) = \langle \boldsymbol{\Phi}_{\mathcal{X}}(\mathbf{x}), \boldsymbol{\Phi}_{\mathcal{Y}}(\mathbf{y}) \rangle_{\mathcal{F}_{\mathcal{X}} \times \mathcal{F}_{\mathcal{Y}}} = \langle \mathbf{f}_{\boldsymbol{\Phi}_{\mathcal{X}}(x)}, \mathbf{g}_{\boldsymbol{\Phi}_{\mathcal{Y}}(y)} \rangle_{\mathcal{B}_{\mathcal{X}} \times \mathcal{B}_{\mathcal{Y}}} = \exp\left[\frac{1}{\sqrt{d_k}}(\mathbf{W}_Q\mathbf{x})^T(\mathbf{W}_K\mathbf{y})\right]$$

From the representational perspective, it is sufficient to know the bilinear (reproducing) form of the kernel above in the MHN updates without requiring an explicit computation of $\{\boldsymbol{\Phi}_{\mathcal{X}(\mathbf{x})}, \boldsymbol{\Phi}_{\mathcal{Y}(\mathbf{y})}\}$ or sampling from the Banach spaces, $\mathcal{F}_{\mathcal{X}} : \mathbf{W}_Q\mathbf{x}, \mathcal{F}_{\mathcal{Y}} : \mathbf{W}_K\mathbf{y}$, or estimation of the manifold functional forms $\{\mathcal{B}_{\mathcal{X}}, \mathcal{B}_{\mathcal{Y}}\}$. Mathematically, these constructions can be formalized as:

$$\mathcal{B}_{\mathcal{X}} = \{\mathbf{f}_{\mathbf{v}} : \mathcal{X} \to \mathcal{R} : \mathbf{f}_v(\mathbf{x}) = \langle \boldsymbol{\Phi}_{\mathcal{X}(\mathbf{x})}, \mathbf{v} \rangle_{\mathcal{F}_X \times \mathcal{F}_Y}; \mathbf{v} \in \mathcal{F}_{\mathcal{Y}}, \mathbf{x} \in \mathcal{X}\}$$

$$= \left\{ \mathbf{f}_{(\mathbf{W}_K\mathbf{y})}(\mathbf{x}) = \exp\left[\frac{(\mathbf{W}_Q\mathbf{x})^T\mathbf{W}_K\mathbf{y}}{\sqrt{d_k}}\right]; \mathbf{W}_K\mathbf{y} \in \mathcal{F}_{\mathcal{Y}}, \mathbf{x} \in \mathcal{X} \right\}$$

$$\mathcal{B}_{\mathcal{Y}} = \{\mathbf{g}_{\mathbf{u}} : \mathcal{Y} \to \mathcal{R} : \mathbf{g}_u(\mathbf{y}) = \langle \mathbf{u}, \boldsymbol{\Phi}_{\mathcal{Y}(\mathbf{y})} \rangle_{\mathcal{F}_X \times \mathcal{F}_Y}; \mathbf{u} \in \mathcal{F}_{\mathcal{X}}, \mathbf{y} \in \mathcal{Y}\}$$

$$= \left\{ \mathbf{g}_{(\mathbf{W}_Q\mathbf{x})}(\mathbf{y}) = \exp\left[\frac{(\mathbf{W}_Q\mathbf{x})^T\mathbf{W}_K\mathbf{y}}{\sqrt{d_k}}\right]; \mathbf{W}_Q\mathbf{x} \in \mathcal{F}_{\mathcal{X}}, \mathbf{y} \in \mathcal{Y} \right\}$$

By extension, the kernel form for HEN can be formalized as follows:

$$K_{\text{HEN}}(\mathbf{x}, \mathbf{y}) = \exp\left[\beta(\boldsymbol{\Phi}_{\mathbf{enc}}(\mathbf{x}))^T(\boldsymbol{\Phi}_{\mathbf{enc}}(\mathbf{y}))\right]$$

$$\mathcal{B}_{\mathcal{X}} = \left\{ \mathbf{f}_{(\boldsymbol{\Phi}_{\mathbf{enc}}(\mathbf{y}))}(\boldsymbol{\Phi}_{\mathbf{enc}}(\mathbf{x})) = \exp\left[\beta(\boldsymbol{\Phi}_{\mathbf{enc}}(\mathbf{x}))^T\boldsymbol{\Phi}_{\mathbf{enc}}(\mathbf{y})\right]; \boldsymbol{\Phi}_{\mathbf{enc}}(\mathbf{y}) \in \mathcal{F}_{\mathcal{Y}}, \mathbf{x} \in \mathcal{X} \right\}$$

$$\mathcal{B}_{\mathcal{Y}} = \left\{ \mathbf{g}_{(\boldsymbol{\Phi}_{\mathbf{enc}}(\mathbf{x}))}(\boldsymbol{\Phi}_{\mathbf{enc}}(\mathbf{y})) = \exp\left[\beta(\boldsymbol{\Phi}_{\mathbf{enc}}(\mathbf{x}))^T\boldsymbol{\Phi}_{\mathbf{enc}}(\mathbf{y})\right]; \boldsymbol{\Phi}_{\mathbf{enc}}(\mathbf{x}) \in \mathcal{F}_{\mathcal{X}}, \mathbf{y} \in \mathcal{Y} \right\}$$

## 1.2 Eq. (2) is a special case of the Radial Translation-Invariant Exponential Kernel in Eq. (4)

According to the formulation in (Iatropoulos et al., 2022), for continuous valued memories, the suggested form of kernel is a radial translation-invariant exponential kernel with a fixed spatial scale $r$ and parameter $\alpha$ is as follows:

$$K_{(\alpha, r)}(\mathbf{x}, \mathbf{y}) = \exp\left[-\left(\frac{1}{r}||\mathbf{x} - \mathbf{y}||_2\right)^{\alpha}\right]$$

$$= \exp\left[-\left(\frac{1}{r}||\mathbf{x} - \mathbf{y}||_2^2\right)^{\frac{\alpha}{2}}\right]$$

$$= \exp\left[-\left(\frac{1}{r}||\mathbf{x}||_2^2 + ||\mathbf{y}||_2^2 - 2\mathbf{x}^T\mathbf{y}\right)^{\frac{\alpha}{2}}\right]$$

$$= \exp\left[-\left(\frac{1}{r}[||\mathbf{x}||_2^2 + ||\mathbf{y}||_2^2]\right)^{\frac{\alpha}{2}}\right] \cdot \exp\left[\left(\frac{2}{r} \cdot \frac{1}{2} \cdot 2\mathbf{x}^T\mathbf{y}\right)^{\frac{\alpha}{2}}\right]$$

$$\text{Substituting } \alpha = 2, \ \beta = \frac{2}{r} = \frac{1}{\sqrt{d_k}} \ \text{ and } \ ||\mathbf{x}||_2 = ||\mathbf{y}||_2 = 1$$

$$K_{\text{Trans}}(\mathbf{x}, \mathbf{y}) = C \cdot \exp\left[\frac{1}{\sqrt{d_k}}(\mathcal{I}_K\mathbf{x})^T(\mathcal{I}_K\mathbf{y})\right] = C \cdot \exp\left[\beta\mathbf{x}^T\mathbf{y}\right]$$

where $C$ is a constant. This is the form of the transformer kernel corresponding to the update in Eq. (2) and energy functional in Eq. (1) of the main manuscript. We note that spherical normalization was also used in (Ramsauer et al., 2020; Krotov & Hopfield, 2020; Wu et al., 2024) as a mechanism to control the dynamics of the updates to mitigate metastable solutions.

# 2 Ablations on $\beta$ for experiments on choices of $f_{\text{sim}}(\cdot, \cdot)$ and Neural Representation

Table 1: Structural Similarity Index Measure for HEN for increasing $\beta$ values. This table presents the 1-SSIM results (lower is better) for a set of N=6000 images from the MS-COCO dataset. Every row represents the similarity metrics (Dot or $\ell_2$) for five different types of pre-trained encoder-decoder architectures (kl8, vqf8, klf16, vqf16, dVAE). The Image column represents the Modern Hopfield Network formulation, the established baseline model in this comparison. We see that as for increasing temperature values of $\beta$, the recovery stabilizes and produces near exact recoveries for all sets of the trained encoder-decoder architectures. In comparison, we see that the image-based Modern Hopfield network fails to handle a large number of images in memory.

| $\beta$ | Dot-klf8 | Dot-vqf8 | Dot-klf16 | Dot-vqf16 | Dot-Image | Dot-dVAE | L2-klf8 | L2-vqf8 | L2-klf16 | L2-vqf16 | L2-Image | L2-dVAE |
|---|---|---|---|---|---|---|---|---|---|---|---|---|
| 500 | 0.0003 | 0.0213 | 0.0043 | 0.0191 | 0.9089 | 0.0000 | 0.1840 | 0.0523 | 0.2315 | 0.0819 | 0.8815 | 0.0000 |
| 480 | 0.0003 | 0.0213 | 0.0043 | 0.0191 | 0.9089 | 0.0000 | 0.1840 | 0.0523 | 0.2315 | 0.0819 | 0.8815 | 0.0000 |
| 460 | 0.0003 | 0.0213 | 0.0043 | 0.0191 | 0.9089 | 0.0000 | 0.1840 | 0.0523 | 0.2315 | 0.0819 | 0.8815 | 0.0000 |
| 440 | 0.0003 | 0.0213 | 0.0043 | 0.0191 | 0.9089 | 0.0000 | 0.1840 | 0.0523 | 0.2315 | 0.0819 | 0.8815 | 0.0000 |
| 420 | 0.0003 | 0.0213 | 0.0043 | 0.0191 | 0.9089 | 0.0000 | 0.1840 | 0.0523 | 0.2315 | 0.0819 | 0.8815 | 0.0000 |
| 400 | 0.0003 | 0.0213 | 0.0043 | 0.0191 | 0.9089 | 0.0000 | 0.1840 | 0.0523 | 0.2315 | 0.0819 | 0.8815 | 0.0000 |
| 380 | 0.0003 | 0.0213 | 0.0043 | 0.0191 | 0.9089 | 0.0000 | 0.1840 | 0.0523 | 0.2315 | 0.0819 | 0.8815 | 0.0000 |
| 360 | 0.0003 | 0.0213 | 0.0043 | 0.0191 | 0.9089 | 0.0000 | 0.1840 | 0.0523 | 0.2315 | 0.0819 | 0.8815 | 0.0000 |
| 340 | 0.0003 | 0.0213 | 0.0043 | 0.0191 | 0.9089 | 0.0000 | 0.1840 | 0.0523 | 0.2315 | 0.0819 | 0.8815 | 0.0000 |
| 320 | 0.0003 | 0.0213 | 0.0043 | 0.0191 | 0.9089 | 0.0000 | 0.1840 | 0.0523 | 0.2315 | 0.0819 | 0.8815 | 0.0000 |
| 300 | 0.0003 | 0.0213 | 0.0043 | 0.0191 | 0.9089 | 0.0000 | 0.1840 | 0.0523 | 0.2315 | 0.0819 | 0.8815 | 0.0000 |
| 280 | 0.0003 | 0.0213 | 0.0043 | 0.0191 | 0.9089 | 0.0000 | 0.1840 | 0.0523 | 0.2315 | 0.0819 | 0.8815 | 0.0000 |
| 260 | 0.0003 | 0.0213 | 0.0043 | 0.0191 | 0.9089 | 0.0000 | 0.1840 | 0.0523 | 0.2315 | 0.0819 | 0.8815 | 0.0000 |
| 240 | 0.0003 | 0.0213 | 0.0043 | 0.0191 | 0.9090 | 0.0000 | 0.1840 | 0.0523 | 0.2315 | 0.0819 | 0.8815 | 0.0000 |
| 220 | 0.0003 | 0.0213 | 0.0043 | 0.0191 | 0.9090 | 0.0000 | 0.1840 | 0.0523 | 0.2315 | 0.0819 | 0.8815 | 0.0000 |
| 200 | 0.0003 | 0.0213 | 0.0043 | 0.0191 | 0.9499 | 0.0000 | 0.1840 | 0.0523 | 0.2315 | 0.0819 | 0.8815 | 0.0000 |
| 180 | 0.0003 | 0.0213 | 0.0043 | 0.0191 | 0.9503 | 0.0000 | 0.1840 | 0.0523 | 0.2315 | 0.0819 | 0.8815 | 0.0000 |
| 160 | 0.0003 | 0.0213 | 0.0043 | 0.0191 | 0.9506 | 0.0000 | 0.1840 | 0.0523 | 0.2315 | 0.0819 | 0.8815 | 0.0000 |
| 140 | 0.0003 | 0.0213 | 0.0043 | 0.0191 | 0.9514 | 0.0000 | 0.1840 | 0.0523 | 0.2315 | 0.0819 | 0.8815 | 0.0000 |
| 120 | 0.0003 | 0.0213 | 0.0043 | 0.0191 | 0.9521 | 0.0000 | 0.1840 | 0.0523 | 0.2315 | 0.0819 | 0.8815 | 0.0000 |
| 100 | 0.0003 | 0.0213 | 0.6657 | 0.0191 | 0.9518 | 0.0000 | 0.1840 | 0.0523 | 0.2315 | 0.0819 | 0.8815 | 0.0000 |
| 80 | 0.0003 | 0.0213 | 0.6625 | 0.0191 | 0.9517 | 0.0000 | 0.1840 | 0.0523 | 0.2315 | 0.0819 | 0.8815 | 0.0000 |
| 60 | 0.0003 | 0.0213 | 0.6617 | 0.0191 | 0.9501 | 0.7911 | 0.1840 | 0.0523 | 0.2315 | 0.0819 | 0.8815 | 0.0000 |
| 40 | 0.0002 | 0.0213 | 0.6644 | 0.0191 | 0.9481 | 0.7629 | 0.1840 | 0.0523 | 0.2315 | 0.0819 | 0.8815 | 0.0000 |
| 20 | 0.6635 | 0.0213 | 0.6665 | 0.7104 | 0.9464 | 0.7828 | 0.1840 | 0.0523 | 0.2315 | 0.0819 | 0.8815 | 0.0000 |

Table 2: Mean Squared Error (MSE) Measure for HEN for increasing $\beta$ values. This table presents the MSE results (lower is better) for a set of N=6000 images from the MS-COCO dataset. Every row represents the similarity metrics (Dot or $\ell_2$) for five different types of pre-trained encoder-decoder architectures (kl8, vqf8, klf16, vqf16, dVAE). The Image column represents the Modern Hopfield Network formulation, the established baseline model in this comparison. We see that as for increasing temperature values of $\beta$, the recovery stabilizes and produces near exact recoveries for all sets of the trained encoder-decoder architectures. In comparison, we see that the image-based Modern Hopfield network fails to handle a large number of images in memory.

| $\beta$ | Dot-klf8 | Dot-vqf8 | Dot-klf16 | Dot-vqf16 | Dot-Image | Dot-dVAE | L2-klf8 | L2-vqf8 | L2-klf16 | L2-vqf16 | L2-Image | L2-dVAE |
|---|---|---|---|---|---|---|---|---|---|---|---|---|
| 500 | 0.0000 | 0.0005 | 0.0001 | 0.0003 | 0.1788 | 0.0000 | 0.0280 | 0.0057 | 0.0341 | 0.0105 | 0.0906 | 0.0000 |
| 480 | 0.0000 | 0.0005 | 0.0001 | 0.0003 | 0.1788 | 0.0000 | 0.0280 | 0.0057 | 0.0341 | 0.0105 | 0.0906 | 0.0000 |
| 460 | 0.0000 | 0.0005 | 0.0001 | 0.0003 | 0.1788 | 0.0000 | 0.0280 | 0.0057 | 0.0341 | 0.0105 | 0.0906 | 0.0000 |
| 440 | 0.0000 | 0.0005 | 0.0001 | 0.0003 | 0.1788 | 0.0000 | 0.0280 | 0.0057 | 0.0341 | 0.0105 | 0.0906 | 0.0000 |
| 420 | 0.0000 | 0.0005 | 0.0001 | 0.0003 | 0.1788 | 0.0000 | 0.0280 | 0.0057 | 0.0341 | 0.0105 | 0.0906 | 0.0000 |
| 400 | 0.0000 | 0.0005 | 0.0001 | 0.0003 | 0.1788 | 0.0000 | 0.0280 | 0.0057 | 0.0341 | 0.0105 | 0.0906 | 0.0000 |
| 380 | 0.0000 | 0.0005 | 0.0001 | 0.0003 | 0.1788 | 0.0000 | 0.0280 | 0.0057 | 0.0341 | 0.0105 | 0.0906 | 0.0000 |
| 360 | 0.0000 | 0.0005 | 0.0001 | 0.0003 | 0.1788 | 0.0000 | 0.0280 | 0.0057 | 0.0341 | 0.0105 | 0.0906 | 0.0000 |
| 340 | 0.0000 | 0.0005 | 0.0001 | 0.0003 | 0.1788 | 0.0000 | 0.0280 | 0.0057 | 0.0341 | 0.0105 | 0.0906 | 0.0000 |
| 320 | 0.0000 | 0.0005 | 0.0001 | 0.0003 | 0.1788 | 0.0000 | 0.0280 | 0.0057 | 0.0341 | 0.0105 | 0.0906 | 0.0000 |
| 300 | 0.0000 | 0.0005 | 0.0001 | 0.0003 | 0.1788 | 0.0000 | 0.0280 | 0.0057 | 0.0341 | 0.0105 | 0.0906 | 0.0000 |
| 280 | 0.0000 | 0.0005 | 0.0001 | 0.0003 | 0.1787 | 0.0000 | 0.0280 | 0.0057 | 0.0341 | 0.0105 | 0.0906 | 0.0000 |
| 260 | 0.0000 | 0.0005 | 0.0001 | 0.0003 | 0.1787 | 0.0000 | 0.0280 | 0.0057 | 0.0341 | 0.0105 | 0.0906 | 0.0000 |
| 240 | 0.0000 | 0.0005 | 0.0001 | 0.0003 | 0.1787 | 0.0000 | 0.0280 | 0.0057 | 0.0341 | 0.0105 | 0.0906 | 0.0000 |
| 220 | 0.0000 | 0.0005 | 0.0001 | 0.0003 | 0.1786 | 0.0000 | 0.0280 | 0.0057 | 0.0341 | 0.0105 | 0.0906 | 0.0000 |
| 200 | 0.0000 | 0.0005 | 0.0001 | 0.0003 | 0.0913 | 0.0000 | 0.0280 | 0.0057 | 0.0341 | 0.0105 | 0.0906 | 0.0000 |
| 180 | 0.0000 | 0.0005 | 0.0001 | 0.0003 | 0.0899 | 0.0000 | 0.0280 | 0.0057 | 0.0341 | 0.0105 | 0.0906 | 0.0000 |
| 160 | 0.0000 | 0.0005 | 0.0001 | 0.0003 | 0.0833 | 0.0000 | 0.0280 | 0.0057 | 0.0341 | 0.0105 | 0.0906 | 0.0000 |
| 140 | 0.0000 | 0.0005 | 0.0001 | 0.0003 | 0.0824 | 0.0000 | 0.0280 | 0.0057 | 0.0341 | 0.0105 | 0.0906 | 0.0000 |
| 120 | 0.0000 | 0.0005 | 0.0001 | 0.0003 | 0.0815 | 0.0000 | 0.0280 | 0.0057 | 0.0341 | 0.0105 | 0.0906 | 0.0000 |
| 100 | 0.0000 | 0.0005 | 0.1112 | 0.0003 | 0.0758 | 0.0000 | 0.0280 | 0.0057 | 0.0341 | 0.0105 | 0.0906 | 0.0000 |
| 80 | 0.0000 | 0.0005 | 0.1082 | 0.0003 | 0.0746 | 0.0000 | 0.0280 | 0.0057 | 0.0341 | 0.0105 | 0.0906 | 0.0000 |
| 60 | 0.0000 | 0.0005 | 0.1074 | 0.0003 | 0.0698 | 0.4034 | 0.0280 | 0.0057 | 0.0341 | 0.0105 | 0.0906 | 0.0000 |
| 40 | 0.0000 | 0.0005 | 0.1070 | 0.0003 | 0.0659 | 0.3819 | 0.0280 | 0.0057 | 0.0341 | 0.0105 | 0.0906 | 0.0000 |
| 20 | 0.1434 | 0.0005 | 0.0852 | 0.1447 | 0.0635 | 0.3821 | 0.0280 | 0.0057 | 0.0341 | 0.0105 | 0.0906 | 0.0000 |

In this experiment, we examine the impact of the $\beta$ parameter on the performance of HEN in the context of image recovery, the choice of the similarity function $f_{\text{sim}}(\cdot, \cdot)$ in the energy functional.

We explore the influence of various pre-trained neural encoder-decoder architectures on the HEN's ability to converge to stable states and achieve accurate recovery. As in the main manuscript, we evaluated the performance using the structural similarity index measure (1-SSIM) and the Mean squared error (MSE) to assess the image recovery. The experiment was conducted using the MS-COCO dataset subset of 6000 images. This dataset was selected for its diversity and complexity, providing a robust challenge to the storage and recovery capabilities of the Modern Hopfield networks and Hopfield Encoded Networks.

The $\beta$ value was varied from $20 - 500$ in increments of 20. This range was chosen to cover a broad range of the dynamics within the MHN. For the similarity function $f_{\text{sim}}(\cdot, \cdot)$, we examine the $\ell_2$ distance and the dot product. The choices for pre-trained encoder-decoder architectures are the Discrete Variational Autoencoder (D-VAE) and variants of VQ-K8 and VQ-F16. We also compared the performance of the Hopfield encoded networks with that of image-based Modern Hopfield networks that did not utilize any pre-trained encoder-decoder systems. This comparison sheds light on the benefits of incorporating complex encoding mechanisms into Hopfield networks. To maintain consistency in representation, we downsampled the images to a resolution of $28 \times 28 \times 3$.

Tables 1, 2 highlight the critical role of $\beta$ in tuning the network's dynamics, with optimal performance observed beyond specific $\beta$ thresholds. The use of encoded neural representations is stable across a much larger range of $\beta$s. In conjunction with the results on separability of encodings and the results in Section 2.4.1 of main manuscript, this experiment highlights the benefits of leveraging encoded neural representations for addressing practical storage considerations in the MHN framework over alternatives such as kernel based approaches.

In exploring the intricate dynamics of HEN, our investigation does not extend to a universal assertion that any combination of time step $T_f$, update rule/energy function, and well-trained encoder-decoder methods will provide good convergence for all datasets or patterns. The behavior of the update dynamics tends to stabilize after a certain number of iterations ($T_f$), beyond which further improvements are marginal. In our experiments, we found that after $T_f = 100$ iterations, the updates plateau, suggesting that the optimal value of $T_f$ is both dataset-dependent and influenced by factors like data dimensionality and pattern separability.

Additionally, delaying the onset of metastable states is an important feature of our proposed approach, as it enables the system to store more content within encoded representations, potentially leading to a higher memory-capacity ceiling. However, this does not imply that the method is entirely free from metastable states or that it will always guarantee convergence for all possible input distributions.

Certain real-world datasets with high noise, irregular structure, or non-separable patterns may still present challenges, even for our enhanced architecture. Future work could involve extending the analysis of metastable state dynamics across different datasets and exploring further variations of update rules or encoding strategies to improve convergence guarantees across a wider range of practical applications.

## 3   Quantifying Meta-Stable States

Recall that SSIM and MSE metrics used in Fig. 2 in the main manuscript quantify the error in the recovery of the stored image content (Fig. 1, column 4). By definition, meta-stable states correspond to degenerate solutions (among entities in the recovered content) in the energy landscape. To quantify the prevalence of these meta-stable states as a function of the dynamics of the HEN, we report an additional proxy for identifying such behavior.

Specifically, we report the relative rank (RR $= R_{\hat{\mathbf{S}}}/R_{\hat{\boldsymbol{\Xi}}}$) of the recovered state matrix ($\hat{\mathbf{S}} = [\hat{\mathbf{s}}_1, \ldots, \hat{\mathbf{s}}_N]$) to that of the memory bank ($\hat{\boldsymbol{\Xi}} = [\hat{\xi}_1, \ldots, \hat{\xi}_N]$) for various values of $\beta$ during the evolution of the dynamics in Eq. (2). In Fig. 1, we plot the RR on the y axis against the number of iterations on the x axis.

$RR < 1$ upon convergence indicates degeneracy in the recovered solutions. Each dashed line corresponds to different values of the temperature $\beta$. For this illustration, we use the dVAE encoder with the dot product formulation of the HEN and 6000 images from the MS-COCO dataset (i.e. the same setup as the green line in Fig. 2 of the main manuscript). We observe very consistent

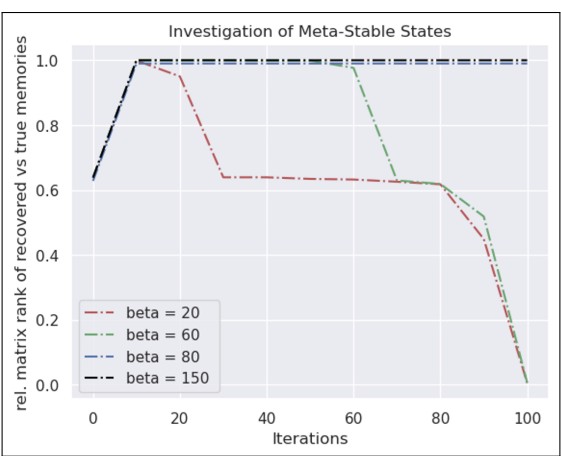

Figure 1: Tracking the evolution of meta-stable states during the dynamics of the HEN. We plot the relative rank (RR $= R_{\hat{\mathbf{S}}}/R_{\hat{\mathbf{\Xi}}}$) of the recovered state matrix ($\hat{\mathbf{S}} = [\hat{\mathbf{s}}_1, \ldots, \hat{\mathbf{s}}_N]$ ) to that of the memory bank ($\hat{\mathbf{\Xi}} = [\hat{\xi}_1, \ldots, \hat{\xi}_N]$). Lower matrix rank signals the presence of meta-stable states. For a judiciously chosen $\beta$, the HEN provides near-perfect recall for this collection of images

trends with those seen in the 1-SSIM and MSE metrics, where near-zero values of these metrics signal no degeneracy (no meta-stable states), while increasing non-zero values indicate imperfect retrieval due to the presence of meta-stable states (such as with row 1, column 4 of Fig. 2). Early in the optimization ($t < 20$ iterations), the $\{\hat{\mathbf{s}}_i^{(t)}\}$ contain information corresponding to encoding the masked image signal. These vectors are distinct across input examples and also exhibit some baseline (non-zero) correlation with the stored memories $\{\hat{\xi}_i\}$ . As the dynamics evolve, we observe that for low values of $\beta = [20, 50]$, the dynamics destabilize to low-rank solutions, collapsing the retrieval fidelity. For sufficiently high $\beta = [80, 150]$, the dynamics stabilize over a period of time, leading to near-perfect recovery ( consistent with $1 - \text{SSIM} = \text{MSE} = 0$)

Finally, the work of (Wu et al., 2024) leverages the equivalence between MHNs, kernel methods, and transformers to directly address the issue of separability of patterns under arbitrary data-distributions. Their framework introduces a second separation maximization objective that is added to the energy minimization objective (i.e. Eq.(1) in the main manuscript). The separation objective is based on a kernel transformation parameterized as a learnable linear kernel or deep network (i.e. encoder) and radial kernel. Thus, pattern retrieval is a two step process that interleaves a stochastic optimization of the encoding parameters with the retrieval dynamics of the Modern Hopfield Network. Here, in each step, local convergence is achieved either as a mini-batch or full batch gradient descent routine. This strategy was shown to provide improvements over the traditional MHNs and their variants upto dataset sizes of 500 for image retrieval. In contrast, our framework fixes the encoding transformation as defined by a pre-trained VAE. As the VAE parameters are frozen apriori, this avoids an expensive outer optimization loop, in turn providing expedient retrieval for datasets of relatively larger size (15000 examples) as well as support hetero-association.

## 4   Handling cross-stimuli using pixelized language-image association

To test language-image associations, we utilized the unique set of captions associated with each image in the COCO dataset. Specifically, we employed Python's `hashlib.sha256(.)` function to hash the captions generating a unique ID text string to associate with the image. Initially, we created a memory bank $\{\hat{\xi}_n^{\mathbf{T}}\}$ by converting the hashed captions into pixelized text representations using a generic text-to-pixel function. Subsequently, both the pixelized text and the corresponding images were processed through the same encoder. The resulting vectors were concatenated to form the elements of the memory bank $\{\hat{\xi}_n = [\hat{\xi}_n^{\mathbf{I}}; \hat{\xi}_n^{\mathbf{T}}]\}$.

During the query phase, we supplied only the pixelized text part of the encoded vector, setting the image component to zero. The Hopfield network iteratively updated the image encoding vector, which was passed through the corresponding decoder to reconstruct the image. Fig. 2 illustrates the

Table 3: The table displays the performance of various encoded cross-modal HEN compared to image-based Modern Hopfield networks as the memory bank $\{\hat{\xi}_n\}$ increases. All of the different encoders performed well. The first line shows the CLIP-encoded cross-modal representations, while the following lines present pixelized text-encoded representations for increasing image sizes.

| NUM IMAGES 1-SSIM, MSE | vqf8 | vqf16 | Image | D-VAE |
|---|---|---|---|---|
| 6000-CLIP | 0.016, 0.000 | 0.024, 0.000 | 0.681, 0.118 | **0.000, 0.000** |
| 6000 | 0.016, 0.000 | 0.024, 0.000 | 0.952, 0.214 | **0.000, 0.000** |
| 8000 | 0.016, 0.000 | 0.023, 0.000 | 0.952, 0.215 | **0.000, 0.000** |
| 10000 | 0.015, 0.000 | 0.024, 0.000 | 0.952, 0.215 | **0.000, 0.000** |
| 15000 | 0.015, 0.000 | 0.024, 0.000 | 0.952, 0.215 | **0.000, 0.000** |

network's progression in reconstructing the image based on the pixelized text input. The recurrent updates in the HEN iteratively reconstructed the full image.

As illustrated in Figs. 2, the HEN network is able to recall perfectly using pixelized cross-stimuli associations. Table 3 (repeated from the main section of the paper for convenience) reveals that all HEN variants with different encodings still outperformed traditional image-based MHNs even as the number of image patterns to store increased. While the HEN can recall accurately based on cross-stimuli associations, we expect such associations to be unique as in the case of stimuli from the same domain/modality (See result in Fig. 6 of the main manuscript).

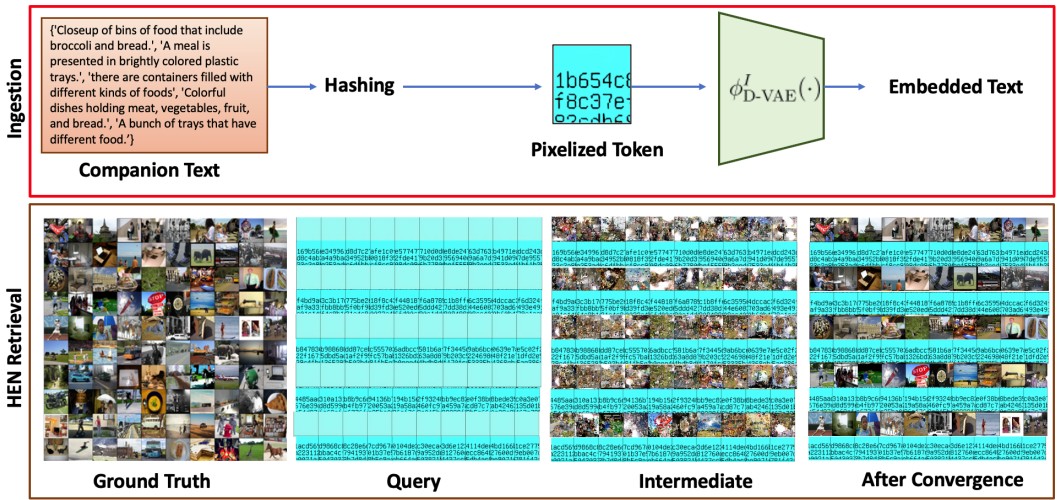

Figure 2: Step-by-step progression of a cross-modal query using only text input. **Top Row:** The companion text is pixelized and encoded (using D-VAE in this example). This encoded representation is used to reconstruct the complete image. **Bottom Row:** The reconstruction process for the experiment in Subsection 4. **(L-R)** Ground Truth, Iteration $t = 0$ starting with a blank canvas with the provided pixelized text inputs as query prompts, an intermediate update, and finally the fully reconstructed image. This visualization effectively demonstrates the network's ability to accurately reconstruct the image from a text-only input modified into an image-based representation