# OpenReview forum: "Modern Hopfield Networks meet Encoded Neural Representations - Addressing Practical Considerations"
_NeurIPS.cc/2024/Workshop/UniReps — UniReps_

### Official Review · Reviewer_H19V · 2024-10-06
**Great extension to Modern Hopfield Networks**

**Rating:** 8
**Confidence:** 3

**Review:**

In this paper, the authors present a method to improve the performance of modern Hopfield networks (MHN) by using variational autoencoders to make the memories stored within the MHN more separable. The authors show that the recall of these memories are improved when the images are encoded by the VAE.  Furthermore, the authors show how different VAEs may be used to process data of different modalities, and then test their model’s capability as a cross-stimuli query retrieval method, showing that it is able to retrieve an associated image with its text query and vice versa, as long as the text-image pairs are unique.

Overall, this is an interesting extension to MHNs, and their extension which helps MHNs process multimodal data is a great contribution to this workshop.

---

### Official Review · Reviewer_fB2d · 2024-10-06
**Modern Hopfield Networks meet Encoded Neural Representations**

**Rating:** 7
**Confidence:** 3

**Review:**

The study introduces "encoded neural representations" inspired by pre-memory encoding in the dentate gyrus, and uses this mechanism to improve the separability of input patterns before training on MHN. The authors evaluated the performance of their model on the MS-COCO dataset, and further demonstrated the capabilities of this framework by learning hetero-associations of image and text. With optimal pairing with encoders, the framework provides near perfect or perfect performance on the dataset.

Overall, the paper provides a novel brain-inspired approach to address the problem of overlapping inputs. To further improve the paper and maximize its application, here are some aspects that the authors may consider focusing on:

1. Quality of the encoding process - this should be addressed to maximize the generalizability of the proposed framework to noisy or complex data.
2. Computational cost of the encoding process - for example, if longer, more implicit input sequences are used instead of short texts or tags. In human memory, fuzzy, implicit descriptors can be used instead of an explicit definition to retrieve multimodal information.
3. Learning other types of associations - the model learns hetero-associations between text and image well, do we expect generations to other types of associations, e.g., images of the same entity from different angles, temporal associations?
4. Stability when using different encodings - although the authors report on the performance of the framework when using different encodings, there's no description of the robustness of the approach when the inputs are poorly encoded.

---

### Official Review · Reviewer_2PPT · 2024-10-07
**The paper introduces Hopfield Encoding Networks (HEN), which integrates encoded neural representations into Modern Hopfield Networks (MHN) to address practical issues like meta-stable states and enhance pattern separability. The proposed framework aims to improve the utility of MHNs in large-scale content storage and retrieval by employing encoded representations, leading to better scalability and fewer spurious states. Experimental results show that HEN enhances pattern separability, supports hetero-association, and enables efficient recall across different modalities.**

**Rating:** 7
**Confidence:** 2

**Review:**

**Quality**:
The paper presents a well-structured framework that effectively integrates Modern Hopfield Networks (MHNs) with encoded neural representations to address known limitations in MHNs, particularly meta-stable states and pattern separability challenges. The experiments are well-designed, using realistic datasets (MS-COCO) and metrics to evaluate performance. The inclusion of hetero-associative memory using cross-modal (text-image) associations demonstrates the broader applicability of the approach.

**Clarity**:
The paper is generally clear and provides comprehensive background information on MHNs, kernel memory networks, and their limitations. The proposed enhancements are explained step-by-step, but some mathematical formulations (such as energy minimization and kernel equivalences) may be challenging for readers without a strong mathematical background. Including more intuitive examples could help in better understanding the core concepts.

**Originality**:
The introduction of encoded neural representations into MHNs to improve pattern separability is a novel contribution. The authors extend MHNs by using a pre-trained encoder-decoder architecture, which results in enhanced retrieval performance and reduction of meta-stable states. The paper also explores the equivalence of MHNs with transformer attention mechanisms, adding to the theoretical depth of the work.

**Significance**:
This work is a significant step toward making MHNs more practical for large-scale associative memory tasks. By leveraging encoded representations, the authors effectively tackle issues that limit the scalability of MHNs. The introduction of hetero-association to enable retrieval with natural language queries is particularly noteworthy, as it expands the usability of MHNs in more diverse applications.

**Strength**:
- Novel Framework: The introduction of Hopfield Encoding Networks (HEN) that integrates encoded neural representations into MHNs is innovative and addresses critical challenges like meta-stable states and poor pattern separability.
- Comprehensive Experiments: Extensive experiments on a realistic dataset (MS-COCO) validate the proposed method, showcasing enhanced recall, storage capacity, and meta-stable state reduction.
- Hetero-associative Capabilities: The paper demonstrates the utility of HEN in hetero-association, allowing cross-modal retrieval, which broadens its practical applications.

**Points of Consideration**:
- Mathematical Complexity: Some of the theoretical aspects, especially related to energy minimization and equivalence with transformers, are presented in a way that might be difficult for readers without a strong background in these topics.
- Scalability Analysis: Although the paper discusses the effectiveness of HEN, a more in-depth discussion of scalability concerning computational complexity and real-time applicability would be valuable.
- Limited Ablation Studies: While different encoders were compared, a more detailed ablation study showing the specific impact of various components of HEN (e.g., encoder architecture choices) could provide further insights into the method's robustness.

**Final Point**:
Overall, this paper contributes strongly to improving Modern Hopfield Networks by integrating encoded representations, leading to increased storage capacity and better recall performance. The approach is validated through experiments that highlight its practical advantages, though the lack of detailed scalability analysis could limit accessibility for some readers.

---

### Author Response · Authors · 2024-10-30
**Thank you for the feedback. We have incorporated the suggestions into our camera ready submission.**

Here are the changes we incorporated into our paper:

- **Scalability Analysis**:  We have added more detail regarding the scalability of the Hopfield Encoding Networks (HEN). Specifically, we enhanced the discussion on the impact of increasing dataset size on retrieval performance, showing that HEN maintains stability even as the memory bank scales from 6,000 to 15,000 images. We addressed how the stable performance was achieved by leveraging pre-trained encoder-decoder architectures, which effectively reduced meta-stable states.

- **Clarification of Ablation Studies**: We revised the presentation of our ablation studies to make it clearer how we evaluated the effects of different encoder-decoder pairs on recovery performance. Specifically, we emphasized the existing analysis involving five different pre-trained architectures across two different (dot product and $\ell_2$ similarity metrics), which demonstrated that HEN maintains stability and robustness across these varied configurations. Additional details are presented in Appendix Section 2, which includes a more comprehensive breakdown of retrieval performance across different parameter choices.

- **Stability with Poorly Encoded Inputs**: We would like to clarify that our original experiments already addressed robustness with poorly encoded or incomplete inputs, particularly in the cross-modal cases. For instance, we used scenarios where there was zero image representation and only text encoding. These experiments, as discussed in the original manuscript, illustrate the resilience of HEN in handling incomplete representations, which aligns with the purpose of associative memory architectures.

- **Learning Other Types of Associations**: Although the current study focuses on cross-modal associations, we acknowledge the potential extension of HEN to other types of associations. To address this, we have included a forward-looking statement in the conclusion suggesting how future work might extend HEN to dynamic and sequential memory tasks, citing relevant earlier work to underscore this potential applicability.

We hope these modifications align well with your feedback, and we appreciate the recognition of our contributions to advancing Modern Hopfield Networks. Thank you for your constructive suggestions.

---

### Decision · Program_Chairs · 2024-10-10

**Decision:**

Accept (Oral)

**Comment:**

In light of the positive reviewers' feedback and relevancy of the submission, we are pleased to accept this paper for presentation at UniReps 2024. We kindly ask the authors to incorporate the reviewers' suggestions and feedback in the final camera-ready version of the manuscript.